# Non-Linear Coordination Graphs

**Yipeng Kang***
Tsinghua University
fringsoo@gmail.com

**Tonghan Wang***
Harvard University
twang1@g.harvard.edu

**Qianlan Yang***
UIUC
qianlan2@illinois.edu

**Xiaoran Wu**
Tsinghua University
wuxr17@tsinghua.org.cn

**Chongjie Zhang**
Tsinghua University
chongjie@tsinghua.edu.cn

## Abstract

Value decomposition multi-agent reinforcement learning methods learn the global value function as a mixing of each agent's individual utility functions. Coordination graphs (CGs) represent a higher-order decomposition by incorporating pairwise payoff functions and thus is supposed to have a more powerful representational capacity. However, CGs decompose the global value function linearly over local value functions, severely limiting the complexity of the value function class that can be represented. In this paper, we propose the first non-linear coordination graph by extending CG value decomposition beyond the linear case. One major challenge is to conduct greedy action selections in this new function class to which commonly adopted DCOP algorithms are no longer applicable. We study how to solve this problem when mixing networks with LeakyReLU activation are used. An enumeration method with a global optimality guarantee is proposed and motivates an efficient iterative optimization method with a local optimality guarantee. We find that our method can achieve superior performance on challenging multi-agent coordination tasks like MACO.

## 1 Introduction

Cooperative multi-agent problems are ubiquitous in real-world applications, such as crewless aerial vehicles [20, 35] and sensor networks [38]. However, learning control policies for such systems remains a major challenge. Recently, value decomposition methods [26] significantly push forward the progress of multi-agent reinforcement learning [22, 24, 28, 29, 30]. In these methods, a centralized action-value function is learned as a mixing of individual utility functions. The mixing function can be conditioned on global states [22] while individual utility functions are based on local action-observation history. The advantage is that agents can utilize global information and avoid learning non-stationarity [27] via centralized training, while maintaining scalable decentralized execution.

Notably, the major focus of the value function decomposition research was on full decomposition, where local utility functions are based on actions and observations of a single agent. Full decomposition leads to several drawbacks, such as miscoordination problems in partially observable environments with stochastic transition functions [28, 32] and a game-theoretical pathology called relative overgeneralization [18, 4]. Relative overgeneralization embodies that, due to the concurrent learning and exploration of other agents, the employed utility function may not be able to express optimal decentralized policies and prefer suboptimal actions that give higher returns on average.

---

*These authors contributed equally to this work.

*Code is available at `https://github.com/fringsoo/CGMIX`

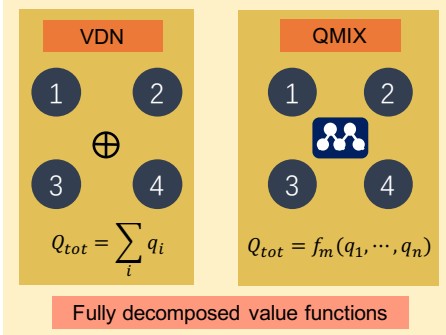
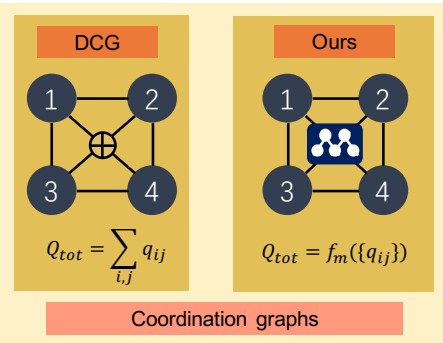

Figure 1: Different value decomposition methods. VDN [26] and QMIX [22] represent the global $Q$-function as a linear and monotonic combination of individual utility functions. Conventional coordination graphs (CGs) [9, 4] learn a linear decomposition of pairwise payoff functions. Our work extends CGs by introducing non-linear combination of payoff functions.

Coordination graph (CG) learning [9] holds the promise to address these problems while preserving the advantages of value decomposition methods. In a CG, each vertex represents an agent, and each (hyper-) edge stands for a payoff function that is defined on the joint action-observation space of the connected agents. The existence of payoff functions increases the granularity of decomposition, and connected agents explicitly coordinate with each other. As a result, a CG represents a higher-order factorization of the global value function and represents a much larger value function class.

However, coordination graphs typically assume a linear decomposition of the value function among sub-groups of agents, which is too simple to represent credit assignment in complex tasks. In this paper, we solve this long-standing problem and extend CG by introducing non-linear and learnable value decomposition. To our best knowledge, it is the first study on non-linear coordination graphs.

The major challenge against extending CGs beyond linear cases is the calculation of value-maximizing actions. When linearly decomposed, DCOP algorithms [6] can find a globally greedy action via message passing. However, when the mixing function is non-linear, DCOP algorithms are no longer applicable. To address this problem, we develop a DCOP method for non-linear mixing functions represented as a deep network with LeakyReLU (or ReLU) activation.

The core idea of our non-linear DCOP algorithm is to exploit the fact that these neural networks are equivalent to piece-wise linear functions. For each linear piece, a value-maximizing action can be found by classic DCOP. However, this action may fall out of the domain of the linear piece and is thus infeasible. We first prove that such a *shifted* action indicates a better solution in its domain. Based on this conclusion, we first show how to find a feasible joint action with the global optimal value, and then derive an iterative algorithm with local optimum convergence guarantee.

We demonstrate the improved representational capacity of our Non-Linear Coordination Graphs (NL-CG) on a matrix game by comparing the learned Q functions to those learned by conventional coordination graphs. We then evaluate our method on the Multi-Agent COordination (MACO) Benchmark [33] for its high requirements on close inter-agent coordination. The experimental results show the superior performance enabled by the non-linear value decomposition.

## 2 Preliminaries

In this paper, we focus on fully cooperative multi-agent tasks that can be modelled as a **Dec-POMDP** [17] consisting of a tuple $G = \langle I, S, A, P, R, \Omega, O, n, \gamma \rangle$, where $I$ is the finite set of $n$ agents, $\gamma \in [0, 1)$ is the discount factor, and $s \in S$ is the true state of the environment. At each timestep, each agent $i$ receives an observation $o_i \in \Omega$ drawn according to the observation function $O(s, i)$ and selects an action $a_i \in A$. Individual actions form a joint action $\boldsymbol{a} \in A^n$, which leads to a next state $s'$ according to the transition function $P(s'|s, \boldsymbol{a})$, a reward $r = R(s, \boldsymbol{a})$ shared by all agents. Each agent has local action-observation history $\tau_i \in T \equiv (\Omega \times A)^* \times \Omega$. Agents learn to collectively maximize the global return $Q_{tot}(s, \boldsymbol{a}) = \mathbb{E}_{s_{0:\infty}, a_{0:\infty}}[\sum_{t=0}^{\infty} \gamma^t R(s_t, \boldsymbol{a}_t)|s_0 = s, \boldsymbol{a}_0 = \boldsymbol{a}]$.

Estimating $Q_{tot}$ is at the core of multi-agent reinforcement learning (MARL) [26, 24, 14, 34, 12]. The complexity of calculating $\max Q_{tot}$ grows exponentially ($|A|^n$) with the number of agents [15, 31]. To solve this problem, value-based MARL decomposes the global action-value function into local utility functions $q_i$ and guarantees that the global maximizer can be obtained locally: $\arg\max_{\boldsymbol{a}} Q_{tot}(s, \boldsymbol{a}) = \left(\arg\max_{a_1} q_1(\tau_1, a_1), \dots, \arg\max_{a_n} q_n(\tau_n, a_n)\right)^{\mathrm{T}}$. Value decomposition network (VDN) [26] satisfies this condition by learning the global value function as a summation of local utilities (Fig. 1-left). QMIX [22] extends the function class of VDN by learning $Q_{tot}$ as a monotonic mixing of local utilities. The mixing function is parameterized so that: $\frac{\partial Q_{tot}(s, \boldsymbol{a})}{\partial q_i(\tau_i, a_i)} \geq 0$.

## 2.1 Coordination Graphs

For fully decomposed value functions, local utility functions are conditioned on local action-observation history. Coordination Graphs (CGs) [9] increase the representational capacity of fully decomposed value functions by introducing higher-order payoff functions. Specifically, a coordination graph is a tuple of a vertex set and an edge set: $\mathcal{G} = \langle \mathcal{V}, \mathcal{E} \rangle$. Each vertex $v_i \in \mathcal{V}$ represents an agent $i$, and (hyper-) edges in $\mathcal{E}$ represent coordination dependencies among agents. In previous work, the global value functions are decomposed linearly based on the graph topology:

$$Q_{tot}(\boldsymbol{\tau}, \boldsymbol{a}) = \frac{1}{|\mathcal{V}|} \sum_i q_i(\tau_i, a_i) + \frac{1}{|\mathcal{E}|} \sum_{\{i,j\} \in \mathcal{E}} q_{ij}(\boldsymbol{\tau}_{ij}, \boldsymbol{a}_{ij}), \tag{1}$$

where $q_i$ and $q_{ij}$ is *utility* functions for individual agents and pairwise *payoff* functions, respectively. $\boldsymbol{\tau}_{ij} = \langle \tau_i, \tau_j \rangle$ and $\boldsymbol{a}_{ij} = \langle a_i, a_j \rangle$ is the joint action-observation history and action of agent $i$ and $j$.

Previous work studies different aspects of such coordination graphs. It is shown that higher-order factorization is important on tackling an exponential number of joint actions [5]. Sparse cooperative Q-learning [11] learns value functions for sparse coordination graphs with pre-defined and static topology. Zhang *et al.* [38] propose to learn minimized dynamic coordination sets for each agent. DCG [4] incorporates deep function approximation and parameter sharing into coordination graphs and scales to large state-action spaces. CASEC [33] and SOP-CG [36] studies how to build sparse deep coordination graphs that are adaptive to the dynamic coordination requirements. In all these works, the global value function is represented as a summation of local value functions. To the best of our knowledge, this paper presents the fist CG learning method with non-linear value decomposition.

## 2.2 Message Passing

Within a coordination graph, the greedy action selection required by Q-learning can not be completed by simply computing the maximum of individual utility and payoff functions. Instead, distributed constraint optimization (DCOP) [6] techniques are used. **Max-Sum** [25] is a popular implementation of DCOP. Max-Sum finds near-optimal actions on a CG $\mathcal{G} = \langle \mathcal{V}, \mathcal{E} \rangle$ via multi-round message passing on a bipartite graph $\mathcal{G}_m = \langle \mathcal{V}_a, \mathcal{V}_q, \mathcal{E}_m \rangle$. Each node $i \in \mathcal{V}_a$ represents an agent, and each node $g \in \mathcal{V}_q$ represents a utility ($q_i$) or payoff ($q_{ij}$) function. Edges in $\mathcal{E}_m$ connect $g$ with the associated agent node(s). Message passing starts with sending messages from node $i \in \mathcal{V}_a$ to node $g \in \mathcal{V}_q$:

$$m_{i \to g}(a_i) = \sum_{h \in \mathcal{F}_i \setminus g} m_{h \to i}(a_i) + c_{ig}, \tag{2}$$

where $\mathcal{F}_i$ is the set of nodes in $\mathcal{V}_q$ connected to node $i$, and $c_{ig}$ is a normalizing factor preventing the value of messages from growing arbitrarily large. Messages are then sent from node $g$ to node $i$:

$$m_{g \to i}(a_i) = \max_{\boldsymbol{a}_g \setminus a_i} [q(\boldsymbol{a}_g) + \sum_{h \in \mathcal{V}_g \setminus i} m_{h \to g}(a_h)], \tag{3}$$

where $\mathcal{V}_g$ is the set of nodes in $\mathcal{V}_a$ connected to node $g$, $\boldsymbol{a}_g = \{a_h | h \in \mathcal{V}_g\}$, $\boldsymbol{a}_g \setminus a_i = \{a_h | h \in \mathcal{V}_g \setminus \{i\}\}$, and $q$ represents utility or payoff functions conditioned on $\boldsymbol{a}_g$. Eq. 2 and 3 make up an iteration of message passing. Each agent $i$ can find its local optimal action by calculating $a_i^* = \arg\max_{a_i} \sum_{h \in \mathcal{F}_i} m_{h \to i}(a_i)$ after several iterations of message passing. Notably, Max-Sum and other DCOP algorithms are only applicable to linearly decomposed value functions.

## 2.3 Piece-Wise Linear Neural Networks

Following the recent decade's success of deep neural networks (DNNs), analysis works have been done trying to explain the mechanism of the DNN black-boxes and assess their function approximation capabilities. One conclusion is that a DNN with piece-wise linear (PWL) activation functions (e.g. ReLU, LeakyReLU, PReLU) is equivalent to a PWL function. This kind of DNNs are called piece-wise linear neural networks (PLNNs) [7]. Early papers [16, 19] assess the expressivity of PLNNs by the amount of linear pieces. [2, 23, 10] give more theoretically grounded results about the upper and lower bounds for the amount of pieces. In this sense, it is shown that increasing the depth of a network can generally be exponentially more valuable than increasing the width [19, 8, 2]. Chu *et al.* [7] propose OPENBOX to compute the mathematically equivalent set of linear pieces, which provides an accurate and consistent interpretation of PLNNs. For neural networks with a single layer of hidden nodes, the problem can be reduced to hyper-plane arrangement [37], and linear functions and their domain can be enumerated efficiently [21, 3].

## 3 Method

Our method extends the representational capability of coordination graphs by introducing non-linear factorization of the global value function. Specifically, for a coordination graph $\mathcal{G} = \langle \mathcal{V}, \mathcal{E} \rangle$ (we study complete coordination graphs in this paper), we decompose the global Q as:

$$Q_{tot}(s, \boldsymbol{a}) = f_n(\boldsymbol{q}_i, \boldsymbol{q}_{ij}), \tag{4}$$

where $\boldsymbol{q}_i$ is the vector of all individual utilities, and $\boldsymbol{q}_{ij}$ is the vector of all pairwise payoffs for edges in $\mathcal{E}$. Similar to DCG [4], we learn a shared utility function $f^v$, parameterized by $\theta^v$, and get the individual utility $q_i(\tau_i, a_i) = f^v(a_i | \tau_i; \theta^v)$. The payoffs are estimated by a shared function $f^e$ parameterized by $\theta^e$: $q_{ij}(\tau_i, \tau_j, a_i, a_j) = f^e(a_i, a_j | \tau_i, \tau_j; \theta^e)$.

Different from conventional CGs (Eq. 1), our $f_n$ is a non-linear mixing network whose parameters are generated by a hypernet $f^h$ conditioned on the global state $s$ and parameterized by $\theta^h$. Our discussion is based on LeakyReLU networks with the slope coefficient $\alpha \in [0, 1]$. For efficiently calculating the maximizer of $Q_{tot}$, we require the weights after the first layer of the mixing network to be non-negative. The reason for this non-negativity constraint will be discussed in Lemma 1 and 2. Such a mixing network is effectively a type of input convex neural networks (ICNN [1]). The non-negativity constraint on parameters is somewhat constraint, but we can use the passthrough trick introduced in Proposition 1 of [1] to maintain substantial representation power of the mixing network.

The whole framework, including the utility function $f^v$, the payoff function $f^e$, and the hypernet $f^h$, is updated by minimizing the TD loss:

$$\mathcal{L}_{\text{TD}}(\theta^v, \theta^e, \theta^h) = \mathbb{E}_{(s, \boldsymbol{a}, r, s') \sim \mathcal{D}} \left[ (Q_{tot}(s, \boldsymbol{a}) - (r + \max_{\boldsymbol{a}'} \hat{Q}_{tot}(s', \boldsymbol{a}')))^2 \right], \tag{5}$$

where $\mathcal{D}$ is the replay buffer, and $\hat{Q}_{tot}$ is a target function whose parameters are periodically copied from the function $Q_{tot}$.

The main challenge left, for both the training and execution phase of deep Q-learning, is to select actions that maximize the global Q value at each time step, $\arg\max_{\boldsymbol{a}} Q_{tot}(\boldsymbol{\tau}, \boldsymbol{a})$. An exact solution to this problem is intractable. Though the mixing network is convex, optimizing its output is still very hard since the inputs are correlated variables, whose simple summation is already a hard one to optimize. In the following sections, we first provide a global optimal algorithm for this problem. The complexity of this algorithm grows polynomially with the number of hidden units in the mixing network. To reduce time complexity, we propose an iterative algorithm with local optimum convergence guarantee.

We use the following **notations**. The input to the mixing network is $\boldsymbol{q} = [\boldsymbol{q}_i \| \boldsymbol{q}_{ij}]$, where $[\cdot \| \cdot]$ means vector concatenation. We use $\boldsymbol{q}(\boldsymbol{a})$ to denote the utilities and payoffs corresponding to action $\boldsymbol{a}$. The mixing network has $L$ LeakyReLU linear layers. $d = |\mathcal{V}| + |\mathcal{E}|$ is the input dimension. $m_i$, $\mathbf{W}_i$, and $\boldsymbol{b}_i \in \mathbb{R}^{m_i}$ are the width, weights, and biases of the $i$th layer. The weights after the first layer are non-negative. The input to the activation units at $i$th layer is $\boldsymbol{z}_i$, and the corresponding output is $\boldsymbol{h}_i = \texttt{LeakyReLU}(\boldsymbol{z}_i) = \boldsymbol{c}_i \circ \boldsymbol{o}_i$, where $\boldsymbol{c}_i$ is the value of LeakyReLU activation ($c = \alpha$ when $z < 0$ and $c = 1$ when $z \geq 0$) and $\circ$ is the element-wise multiplication. We call $\boldsymbol{c} = [\boldsymbol{c}_1 \| \dots \| \boldsymbol{c}_L] \in \{\alpha, 1\}^m$, $m = \sum_i m_i$, a *slope configuration* of the mixing network. Subscripts of $\mathbf{W}_i$, $\boldsymbol{b}_i$, $\boldsymbol{z}_i$, and $\boldsymbol{h}_i$ means index. For example, $z_{ij}$ is the $j$th element of $\boldsymbol{z}_i$, and $\mathbf{W}_{ij}$ is the $j$th row of $\mathbf{W}_i$.

## 3.1 Piece-Wise Optimization

The general idea of our action selection algorithm is to utilize the piece-wise linear property of the LeakyReLU network. Given a slope configuration $c$, the mixing network becomes linear, and we can run a DCOP algorithm to get the corresponding maximizer of $Q_{tot}$. The question is that the obtained solution may be out of the domain of the linear piece. We show that, when the weights after the first layer are non-negative, we can ignore whether the obtained local optimal solution is in the piece's domain, and the global maximizer of $Q_{tot}$ is the best of the local optima.

Formally, there are $2^m$ slope configurations in $\mathcal{C}_{all} = \{c | c \in \{\alpha, 1\}^m\}$, each of which makes the mixing network an affine function in $\mathcal{P}_{all} = \{\rho_1, \rho_2, \ldots, \rho_{2^m}\}$. Each affine function $\rho_k$ has a corresponding cell $P_k$ where $q \in P_k$ yields $c^k$ in the forward pass. For each $c^k \in \mathcal{C}_{all}$, by running DCOP, we obtain the local maximum $q_k$ and the corresponding joint action $a_k$ and utilities/payoffs $q_k$. However, it is possible that $q_k$ falls out of $P_k$ and indeed yields another slope configuration $c^{r \neq k}$. We first show that such a *shifted* solution indicates an equal or better solution in its domain.

**Lemma 1.** *Denote affine function pieces and their cells of a fully-connected feedforward mixing network with LeakyReLU activation as $\mathcal{P}_{all} = \{\rho_j\}_1^{2^m}$ and $\{P_j\}_1^{2^m}$. For $q$ in the cell of the $r$th piece, $P_r$, and $\forall \rho_s \in \mathcal{P}_{all}$, we have $\rho_r(q) \geq \rho_s(q)$.*

*Proof.* We start with the *first* difference between $c^r$ and $c^s$. We denote it as $c_{ij}^r \neq c_{ij}^s$. Because this is the first difference, the input to this unit, $z_{ij}$, is the same. Since $q$ is in $P_r$, we have that $c_{ij}^r$ is 1 when $z_{ij}$ is positive and is $\alpha < 1$ when $z_{ij}$ is negative. Therefore, the output satisfies that

$$h_{ij}^r = c_{ij}^r z_{ij} \geq c_{ij}^s z_{ij} = h_{ij}^s. \tag{6}$$

It follows that

$$z_{i+1}^r = \mathbf{W}_{i+1}^{\mathrm{T}} h_i^r + b_{i+1} \geq \mathbf{W}_{i+1}^{\mathrm{T}} h_i^s + b_{i+1} = z_{i+1}^s, \tag{7}$$

because $\mathbf{W}_{i+1} \geq 0$. Other differences in the $i$th layer will lead to the same conclusion. At layer $i+1$, we have

$$h_{i+1}^r = c_{i+1}^r \circ z_{i+1}^r \geq c_{i+1}^s \circ z_{i+1}^r \geq c_{i+1}^s \circ z_{i+1}^s = h_{i+1}^s. \tag{8}$$

The first inequity is because $q$ is in the cell of $\rho_r$. The second inequity is because of Eq. 7 and that LeakyReLU is a monotonic increasing function. The proof can repeat for the following layers and thus holds for the last layer, *i.e.*, $\rho_r(q) \geq \rho_s(q)$. □

Based on Lemma 1, when the local optimal solution, $a_k$, of piece $k$ returned by DCOP actually falls in cell $P_{r \neq k}$, we have $\rho_r(q(a_k)) \geq \rho_k(q(a_k))$. This indicates that, on piece $r$, we will get a solution at least as good as the one on piece $k$. Based on this conclusion, we now prove that the maximum of each piece's local optimum is the global optimum.

**Lemma 2.** *Running DCOP algorithm for each linear piece in $\mathcal{P}_{all}$, then take the maximum value among these pieces, we can get the global optimizer of $Q_{tot}$.*

*Proof.* We have that $\max_q f_n(q) = \max_q \max_{\rho_p \in \mathcal{P}_{all}} \rho_p(q) = \max_{\rho_p \in \mathcal{P}_{all}} \max_q \rho_p(q)$ ($f_n$ is the mixing function, Eq. 4), which means the maximum of local optimal values is the global optimal value. Suppose the global optimal value $Q_{max}$ is found on piece $r$ and the corresponding action is $a_r$, the question is whether $q(a_r)$ is feasible, *i.e.*, $q(a_r) \in P_r$. Assume that $q(a_r)$ falls in $P_{s \neq r}$. Then for piece $s$, $a_r$ is a feasible solution with the value $Q_{max}$ (due to Lemma 1). This means the global optimal solution is always a feasible solution. □

According to Lemma 2, we can run message passing on each piece (Algorithm **??** in Appx. **??**) and then take the best of these results to get the global optimum. Detailed process can be found in Algorithm 1.

The problem of Algorithm 1 is time complexity. When $m$ is small, we enumerate all $2^m$ slope configurations. For a large $m$, we can use the hyperplane arrangement algorithm [7] to calculate exact linear pieces and their domains. Specifically, there are

$$n_{m,d} = \sum_{i=0}^{d} \binom{m}{d-i} \tag{9}$$

pieces need enumerating, where $d$ is the input length. This number is exponential to $d$. To reduce time complexity, we propose an iterative optimization method in the next section.

## 3.2 Iterative Optimization with Local Optimum Guarantee

When $m$ is large, enumerating either $2^m$ or $n_{m,d}$ pieces can be costly. We thus propose an iterative algorithm with *local optimum guarantee* (Algorithm 2) to get approximate solutions. We begin with the slope configuration where all hidden units are activated ($\boldsymbol{c} = \{1\}^m$). We run message passing on the current configuration $\boldsymbol{c}_p$ and get a local optimum solution $\boldsymbol{a}_p$. If the real configuration $\boldsymbol{c}_{real}$ of $\boldsymbol{a}_p$ is not $\boldsymbol{c}_p$, we calculate the local optimum of $\boldsymbol{c}_{real}$. The iteration continues until $\boldsymbol{c}_{real} = \boldsymbol{c}_p$.

Lemma 1 guarantees that the solution values in the iteration are monotonically increasing. Since there is a finite number of configurations, our algorithm can converge to a local optimum.

With a very low probability, a loop would appear in the iteration. The appearance of a loop means that the equity in Lemma 1 holds. In this case, the solutions on the loop have the same value, thus we can stop our iteration when encountering a slope configuration that has been searched.

---

**Algorithm 1** ENUMERATE-OPTIMIZATION
/*Show the case for a two-layer mixing network, but can be easily extended to more layers.*/

    **Input:** $\boldsymbol{f}^{\text{V}} \in \mathbb{R}^{|\mathcal{V}| \times A}$, $\boldsymbol{f}^{\text{E}} \in \mathbb{R}^{|\mathcal{E}| \times A \times A}$, $\mathbf{W}_0$, $b_0$, $\mathbf{W}_1$, $b_1$
    $q_{\max} := -\infty$; $\boldsymbol{a}_{\max} := [\,]$
    /*Initialize the best found solution.*/
    **for** $\boldsymbol{c}_p \in \{\alpha, 1\}^m$ **do**
      /*Calculate the equivalent weights and biases.*/
      $\boldsymbol{W}_{\rho_p} := \mathbf{W}_0 \cdot (\boldsymbol{c}_p \circ \boldsymbol{W}_1)$
      $b_{\rho_p} := (\boldsymbol{c}_p \circ \boldsymbol{W}_1) \cdot b_0^T + b1$
      $q, \boldsymbol{a}, \cdot \leftarrow$
      $w\text{-MAX-SUM}(\boldsymbol{f}^{\text{V}}, \boldsymbol{f}^{\text{E}}, \boldsymbol{W}_{\rho_p}, b_{\rho_p})$
      **if** $q > q_{\max}$ **then**
        $\boldsymbol{a}_{\max} \leftarrow \boldsymbol{a}$
        $q_{\max} \leftarrow q$
        /*Remember only the best actions.*/
      **end if**
    **end for**
    **return** $\boldsymbol{a}_{\max}$
/*Return the selected joint action $\boldsymbol{a}_{\max}$.*/

**Algorithm 2** ITERATIVE-OPTIMIZATION
/*Show the case for a two-layer mixing network, but can be easily extended to more layers.*/

    **Input:** $\boldsymbol{f}^{\text{V}} \in \mathbb{R}^{|\mathcal{V}| \times A}$, $\boldsymbol{f}^{\text{E}} \in \mathbb{R}^{|\mathcal{E}| \times A \times A}$, $\mathbf{W}_0$, $b_0$, $\mathbf{W}_1$, $b_1$, $n_{max}$, $\epsilon$
    $q_{\max} := -\infty$, $\boldsymbol{a}_{\max} := [\,]$, $\boldsymbol{c}_p := \{1\}^m$
    **for** $n \in \{1, \ldots, n_{max}\}$ **do**
      /*Try one configuration at a time.*/
      $\boldsymbol{W}_{\rho_p} := \mathbf{W}_0 \cdot (\boldsymbol{c}_p \circ \boldsymbol{W}_1)$
      $b_{\rho_p} := (\boldsymbol{c}_p \circ \boldsymbol{W}_1) \cdot b_0^T + b1$
      $q, \boldsymbol{a}, \cdot \leftarrow$
      $w\text{-MAX-SUM}(\boldsymbol{f}^{\text{V}}, \boldsymbol{f}^{\text{E}}, \boldsymbol{W}_{\rho_p}, b_{\rho_p})$
      **if** $q > q_{\max}$ **then**
        $\boldsymbol{a}_{\max} \leftarrow \boldsymbol{a}$
        $q_{\max} \leftarrow q$
        $\boldsymbol{c}_{real} \leftarrow$ The real LeakyReLU slope configuration
      **end if**
      **if** $\boldsymbol{c}_{real} \neq \boldsymbol{c}_p$ **then** $\boldsymbol{c}_p \leftarrow \boldsymbol{c}_{real}$ **else** with prob. $\epsilon$ **break** or continue with an unvisited $\boldsymbol{c}_p$.
    **end for**
    **return** $\boldsymbol{a}_{\max}$

---

To increase the possibility of finding a global optimum, we can introduce a simulated annealing mechanism. Each time we find a better solution, with a probability of $1 - \epsilon$, we move to the corresponding piece, and with a probability of $\epsilon$, we jump to a random piece that has not been searched. $\epsilon$ decreases with the iteration number.

**Discussion about loopy graph topology** In this paper, we consider complete graphs when studying non-linear coordination graphs. A concern is that message passing algorithms like Max-Sum may not converge to the optimal solutions in loopy graphs and has an error rate of $e$. Lemma 1 is not affected because it is a property of LeakyReLU Networks. For Lemma 2, the maximum of solutions found by message passing in all slope configurations is the global optimum with a probability of $1 - e$. An error occurs when message passing cannot find the right solution on the piece where the global optimum is located. Our iterative method may stop earlier when message passing returns a wrong solution located in the current cell. The probability of this situation is less than $e$. Thus we have at least a probability of $(1 - e)^n$ ($n$ is the number of iterations) to find the piece where the local optimum is located, and the final probability of finding the local optimum is larger than $(1 - e)^{n+1}$.

## 4 Representational Capability

In this section, we compare the representational capacity of our model against conventional coordination graphs. The comparison is carried out on a two-step cooperative matrix game with four players and two actions. At the first step, Agent 1's action decides which of the two matrix games (Table. 1)

to play in the next timestep. In the second step, the number of agents taking Action B determines the global reward received by the agent team (Table. 1).

| State 2A | | | | | | | State 2B | | | | | |
|---|---|---|---|---|---|---|---|---|---|---|---|---|
| # Action B | 0 | 1 | 2 | 3 | 4 | | # Action B | 0 | 1 | 2 | 3 | 4 |
| Reward | 7 | 7 | 7 | 7 | 7 | | Reward | 0 | -0.1 | 0.1 | 0.3 | 8 |

Table 1: Payoff matrices of the two-step game after Agent 1 chooses the first action. Action A takes the agents to State 2A, while action B takes them to State 2B. The team reward depends on the number of agents taking Action B.

We first theoretically prove that the representational capacity of conventional coordination graphs is unable to represent the $Q$-function of this game. Since the reward is invariant to the identity of agents, *i.e.*, $R(a_1, a_2, a_3, a_4) = R(p(a_1, a_2, a_3, a_4))$, where $p$ is an arbitrary permutation, the learned action-value function should also be permutation invariant. Therefore we can ignore the order of actions in value functions. Now let's focus on State 2B ($s_{2B}$). Conventional coordination graphs need to solve a linear system consisting of five unknowns and five equations. However, for this system, the rank of the augmented matrix (4) is greater than the rank of the coefficient matrix (3). Details can be found in Appx. **??**. Therefore, this system has no solution, and it is impossible for conventional CGs to learn a correct $Q$ function.

We then empirically demonstrate our idea. We train NL-CG and DCG on the task for 5000 episodes under full exploration ($\epsilon = 1$) and examine the learned value functions. Full exploration ensures that both methods explore all state-action pairs. In such a case, the representational capacity of the action-value function approximator remains the only challenge of learning accurate Q functions. For our algorithm, the utility and payoff function is fully connected networks with a single hidden layer of 64 units with a ReLU non-linearity. $\gamma$ is 0.99, and the replay buffer stores the last 500 episodes, from which we uniformly sample batches of size 32 for training. The target network is updated every 100 episodes. The learning rate of RMSprop is set to $5 \times 10^{-4}$. Agents receive the full state as observation, which is represented as an one-hot vector.

Table 2 and 3 show the learned $Q$ by DCG and NL-CG. In line with our theoretical analysis, we can see that DCG learns a sub-optimal strategy of selecting Action A in the first step. By contrast, our method learns the accurate value of Action B in State A and get the optimal strategy. Furthermore, the Q values for State 2B learned by NL-CG is more accurate than those learned by DCG. These results demonstrate that NL-CG's higher representational capacity allows it to accurately estimate the value function of this game whereas DCG cannot. We also note that such an example is common for task with few actions because there are typically more equations than unknowns.

| State A | | | | State 2B | | | | | |
|---|---|---|---|---|---|---|---|---|---|
| Action | A | B | | # Action B | 0 | 1 | 2 | 3 | 4 |
| Q | 6.91 | 5.85 | | Q | 1.32 | -0.89 | -0.77 | 1.41 | 5.89 |

Table 2: Q-functions learned by DCG for the matrix game.

| State A | | | | State 2B | | | | | |
|---|---|---|---|---|---|---|---|---|---|
| Action | A | B | | # Action B | 0 | 1 | 2 | 3 | 4 |
| Q | 6.92 | 7.95 | | Q | 0.19 | -0.12 | 0.20 | 0.63 | 8.02 |

Table 3: Q-functions learned by our method for the matrix game.

## 5 Experiments

In this section, we conduct experiments to show the effectiveness of our method on complex tasks. We benchmark our method on the Multi-Agent COordination (MACO) benchmark [33], which covers various classic coordination tasks in the literature of multi-agent learning and increases their complexity to better evaluate the performance of different algorithms. The MACO benchmark is characterized by a high demand on the sophistication of agent coordination. For NL-CG, we use a mixing network that has one hidden layer with different widths. Detailed hyperparameter settings of our method can be found in Appendix **??**. For fair comparison, we run all our experiments with 5 random seeds and show the mean performance with a $95\%$ confidence interval.

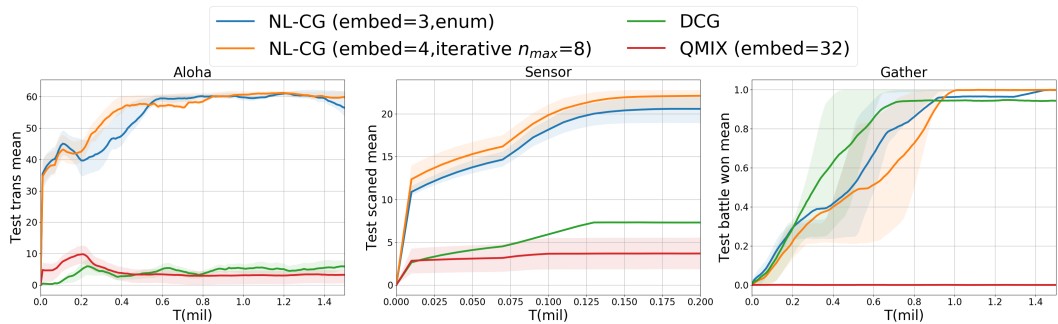

Figure 2: Performance comparison against baselines on the MACO benchmark.

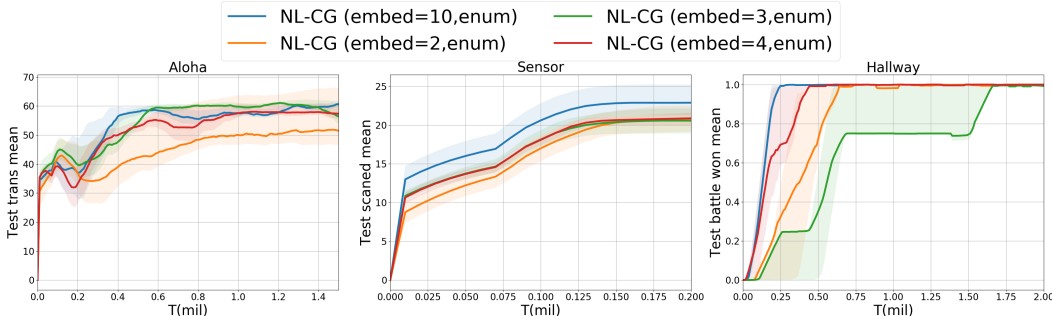

Figure 3: Influence of the size of the mixing network. When the width of the hidden layer is 10, enumerating all linear pieces is quite time-consuming. We thus stop training when we observe its performance reaching or surpassing the best performance achieved by other algorithms.

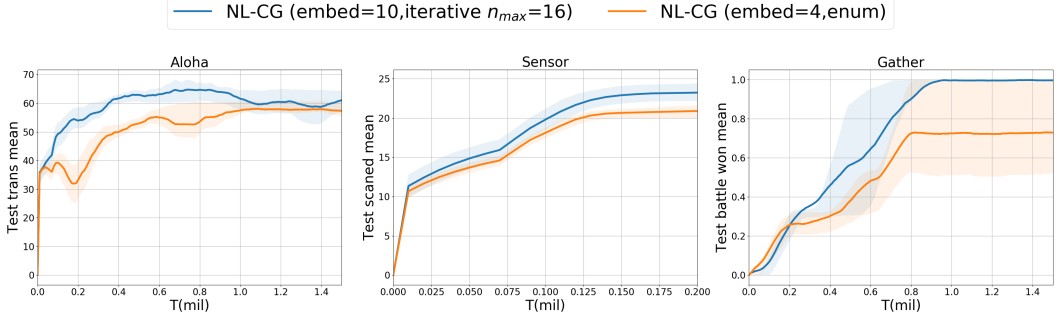

Figure 4: Our iterative optimization method reduces NL-CG's time cost and thus can use a larger mixing network, leading to better performance when checking the same number of linear pieces.

In Fig. 2, we compare NL-CG against the state-of-the-art coordination graph learning method (DCG [4]) and fully decomposed methods (QMIX [22], DICG [13]). For both DCG and our method, we use the complete graphs for all experiments in the paper. For NL-CG, we (1) set the hidden width to $3$ and enumerate all configurations and (2) set the hidden width to $4$ and run our iterative method with $n_{max}=4$. We stop iteration when $n_{max}$ slope configurations are visited. The result shows that our algorithm can outperform conventional CGs significantly. Moreover, our iterative optimization method has comparable performance with the enumeration method, showing its effectiveness. QMIX struggles on these tasks, indicating that these tasks are beyond the representation capacity of a fully decomposed function.

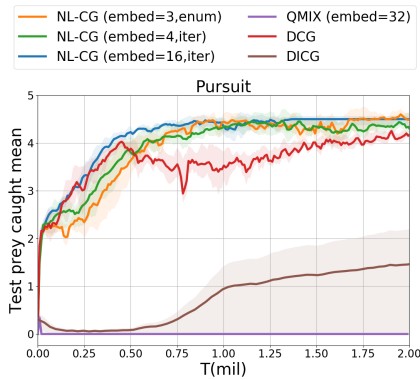

Figure 5: Performance on `Pursuit`.

We further investigate the the influence of the mixing network's width. Specifically, we test NL-CG with a width of 2, 3, 4, and 10 and compare their performance in Fig. 3. It can be observed that, generally, more units (10) in the hidden layer lead to better or at least equal performance than other configurations. This result is in line with our motivation: a powerful non-linear mixing network increases the capability of CGs.

In Fig. 4, we compare our enumerative and iterative methods. These two methods check the same number (16) of linear pieces, but the iterative method can use a hidden layer of 10. As a result, the iterative method outperforms the enumerative method.

Additionally, in Fig. 5, we compare NL-CG against DCG [4], QMIX [22], and DICG [13] on `Pursuit`. We find NL-CG generally outperforms previous methods.

## 5.1 Analysis of the optimality and efficiency of the iterative optimization method

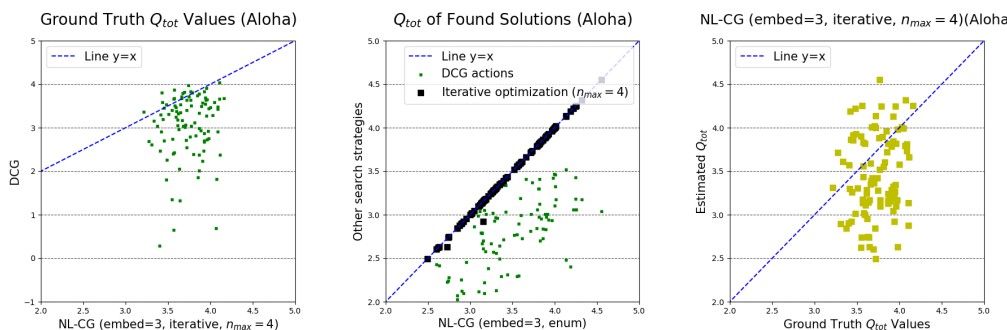

Figure 6: **Left**: Ground-truth Q value comparisons show that NL-CG learns a policy with higher value than DCG. **Middle**: Actions obtained by enumerative and iterative methods have similar values. **Right**: $Q_{tot}$ output of NL-CG against ground-truth Q value (averaged Monte Carlo returns).

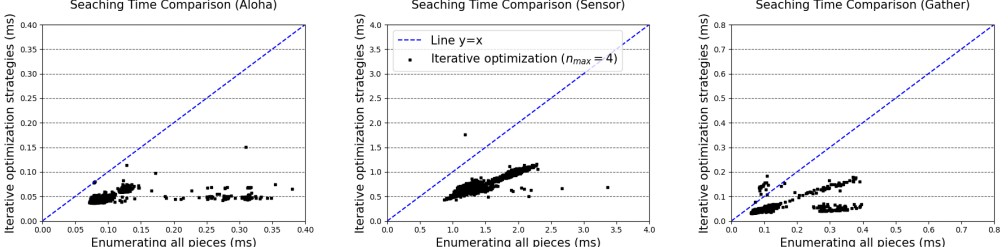

Figure 7: Efficiency of the iterative method. **x-axis**: Time spent by enumerating all pieces. **y-axis**: Time spend by the iterative method. The farther down the line $y = x$, the faster the method is.

Although we have checked the performance of our iterative optimization method, learning curves can not fully reveal its optimality and efficiency. In this section, we provide optimality and efficiency analyses by checking the action selection results and time costs in detail.

In Fig. 6-middle, we compare the $Q_{tot}$ value of our enumerative and iterative optimization methods. We can see that the iterative method is near-optimal after checking only 4 (embed=3) slope configurations: the Q-values of its actions are very close to those selected by enumeration and are better than those selected by DCG. In Fig. 6-left, we compare the ground truth Q estimates of DCG and NL-CG (embed=3, iterative, $n_{max} = 4$). The result shows that NL-CG learns a policy with higher value. In Fig. 6-right, we compare the $Q_{tot}$ values estimated by NL-CG (embed=3, iterative, $n_{max} = 4$) against ground truth Q estimates. The estimation errors on all tested state-action pairs are less than 20%.

In Fig. 7, we compare the time spent by enumerative and iterative optimization methods. It can be found that the iterative method saves 50% to 65% of running time. We can thus conclude that the iterative optimization method provides a good trade-off between complexity and optimality.

## 6   Conclusion

In this paper, we extend coordination graphs beyond linear decomposition by introducing non-linear mixing networks. Experiments manifest its superior representation power on complex tasks that conventional CGs are not able to solve. An important research direction is the stability of non-linear CGs and to get rid of the non-negative constraint on weights of the mixing network. The authors do not see obvious negative societal impacts of our method.

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
