# A Hyperparameters and Infrastructure

When comparing the performance of NL-CG against DCG [4] and QMIX [22] on the MACO benchmark, we adopted the following hyper-parameter settings. For all tasks, we used a discount factor $\gamma = 0.99$ and $\epsilon$-greedy exploration, where $\epsilon$ was linearly decayed from 1 to 0.05 within the first 50,000 training time steps. The replay buffer stored the last 5000 episodes, from which we uniformly sampled batches of size 32 for training. The target network was updated every 200 episodes. The learning rate of RMSprop was set to $5 \times 10^{-4}$, except for the `Pursuit` experiment, which used $1 \times 10^{-2}$ for faster convergence. Every 10000 time steps we paused training and evaluated the model with 300 greedy test trajectories sampled with $\epsilon = 0$.

Each agent processed its local action-observation history at each time step using a linear layer of 64 neurons, followed by a ReLU activation, a GRU of the same dimensionality, and finally a linear layer of $|A|$ neurons. The output served as an individual feature vector and was fed into the utility and payoff function. The parameters of the utility and payoff function were shared amongst agents, who were identified by a one-hot encoded ID in the input. All message passing procedures in NL-CG iterated for 4 rounds. The weights and bias of the mixing network in NL-CG were generated by a 2-layer hyper-network with a hidden layer of 64 ReLU neurons, except for the `Aloha` experiment, which uses one linear layer.

Our implementation of NL-CG was based on the PyMARL[*] [23] framework. We used NVIDIA GeForce RTX 3090 GPUs for training and evaluation.

# B Weighted Max-Sum

Algorithm 1 and Algorithm 2 find global and approximate greedy actions for a non-linear coordination graph, respectively. Both of these two algorithms rely on the weighted Max-Sum algorithm to find local optimal action on a linear piece. We present the weighted Max-Sum algorithm in Algorithm 3.

# C Representational Capability

In Sec. 4, we use a matrix game to show the representational capability of non-linear coordination graphs. For conventional coordination graphs on State 2B of this game, since the value function should be permutation invariant, there are five unknowns $q_1 = q_i(s_{2B}, A)$, $q_2 = q_i(s_{2B}, B)$, $q_3 = q_{ij}(s_{2B}, AA)$, $q_4 = q_{ij}(s_{2B}, AB)$, and $q_5 = q_{ij}(s_{2B}, BB)$ and five equations:

$$\begin{cases} 4q_1 & + 6q_3 & = 0 \\ 3q_1 + & q_2 + 3q_3 + 3q_4 & = -0.1 \\ 2q_1 + 2q_2 + & q_3 + 4q_4 + & q_5 = 0.1 \\ q_1 + 3q_2 & + 3q_4 + 3q_5 = 0.3 \\ & 4q_2 & + 6q_5 = 8 \end{cases} \tag{10}$$

The augmented matrix of this system has a higher rank (4) than its coefficient matrix (3). Therefore, this linear system does not have a solution, which means that a conventional coordination graph cannot represent the accurate value function for this task.

---

[*] `https://github.com/oxwhirl/pymarl`

**Algorithm 3** $w$-MAX-SUM

/*Greedy action selection with $k$ message passing for one linear piece of the mixing network $f_m$. This algorithm is called by Algorithm 1 and 2, and the definition of inputs can be found there.*/

**Input:** $\boldsymbol{f}^{\mathrm{V}} \in \mathbb{R}^{|\mathcal{V}| \times A}$, $\boldsymbol{f}^{\mathrm{E}} \in \mathbb{R}^{|\mathcal{E}| \times A \times A}$, $(\boldsymbol{W}_{\mathcal{V}}, \boldsymbol{W}_{\mathcal{E}}) \in \mathbb{R}^{|\mathcal{V}| + |\mathcal{E}|}$, $b \in \mathbb{R}$

$\boldsymbol{f}^{\mathrm{V}} := \boldsymbol{W}_{\mathcal{V}} \circ \boldsymbol{f}^{\mathrm{V}}$

$\boldsymbol{f}^{\mathrm{E}} := \boldsymbol{W}_{\mathcal{E}} \circ \boldsymbol{f}^{\mathrm{E}}$

$\boldsymbol{\mu}^0, \bar{\boldsymbol{\mu}}^0 := \boldsymbol{0} \in \mathbb{R}^{|\mathcal{E}| \times A}$

/*Initialize forward messages ($\boldsymbol{\mu}$) and backward messages ($\bar{\boldsymbol{\mu}}$).*/

$\boldsymbol{q}^0 := \boldsymbol{f}^{\mathrm{V}}$

/*Initial "Q-value".*/

$q_{\max} := -\infty; \boldsymbol{a}_{\max} := \left[ \arg\max_{a \in \mathcal{A}^i} q^0_{ai} \,\middle|\, i \in \mathcal{V} \right]$

/*Initialize the best found solution.*/

**for** $t \in \{1, \dots, k\}$ **do**

  /*$k$ rounds of message passing.*/

  **for** $e = (i, j) \in \mathcal{E}$ **do**

    /*Update forward and backward messages. Subscripts of $\boldsymbol{f}, \boldsymbol{q}, \boldsymbol{\mu}$ mean indexing.*/

    $\boldsymbol{\mu}^t_e := \max_{a \in \mathcal{A}^i} \left\{ (q^{t-1}_{ia} - \bar{\mu}^{t-1}_{ea}) + \boldsymbol{f}^{\mathrm{E}}_{ea} \right\}$

    /*Forward: maximize sender.*/

    $\bar{\boldsymbol{\mu}}^t_e := \max_{a \in \mathcal{A}^j} \left\{ (q^{t-1}_{ja} - \mu^{t-1}_{ea}) + (\boldsymbol{f}^{\mathrm{E}}_e)^\top_a \right\}$

    /*Backward: maximize receiver.*/

    $\boldsymbol{\mu}^{\boldsymbol{t}}_{\boldsymbol{e}} \leftarrow \boldsymbol{\mu}^{\boldsymbol{t}}_{\boldsymbol{e}} - \frac{1}{|\mathcal{A}^j|} \sum_{a \in \mathcal{A}^j} \mu^t_{ea}$

    /*Normalize forward messages to ensure converging.*/

    $\bar{\mu}^t_e \leftarrow \bar{\mu}^t_e - \frac{1}{|\mathcal{A}^i|} \sum_{a \in \mathcal{A}^i} \bar{\mu}^t_{ea}$

    /*Normalize backward messages to ensure converging.*/

  **end for**

  **for** $i \in \mathcal{V}$ **do**

    /*Update "Q-value" with messages.*/

    $\boldsymbol{q}^t_i := \boldsymbol{f}^{\mathrm{V}}_i + \sum_{e=(\cdot,i) \in \mathcal{E}} \boldsymbol{\mu}^t_e + \sum_{e=(i,\cdot) \in \mathcal{E}} \bar{\boldsymbol{\mu}}^t_e$

    /*Utility plus incoming messages.*/

    $a^t_i := \arg\max_{a \in \mathcal{A}^i} \{ q^t_{ia} \}$

    /*Select the greedy action of agent $i$.*/

  **end for**

  $q' \leftarrow \sum_{i=1}^{|\mathcal{V}|} \boldsymbol{f}^{\mathrm{V}}_{a^i} + \sum_{(i,j) \in \mathcal{E}} \boldsymbol{f}^{\mathrm{E}}_{a^i a^j} + b$

  /*Get the Q-value of the greedy action.*/

  **if** $q' > q_{\max}$ **then**

    $\{ \boldsymbol{a}_{\max} \leftarrow \boldsymbol{a}^t; \ q_{\max} \leftarrow q' \ \boldsymbol{q} \leftarrow \boldsymbol{q}^t_i \}$

    /*Remember the best action.*/

  **end if**

**end for**

**return** $q_{\max}, \boldsymbol{a}_{\max} \in \mathcal{A}^1 \times \dots \times \mathcal{A}^{|\mathcal{V}|}, \boldsymbol{q}$

/*Return the maximum $Q$ value, the corresponding action, and utilities/payoffs.*/