# OpenReview forum: "Non-Linear Coordination Graphs"
_NeurIPS.cc/2022/Conference — NeurIPS 2022 Accept_

### Official Review · Reviewer_Gt8B · 2022-06-25

**Rating:** 7
**Confidence:** 5
**Soundness:** 3 good
**Presentation:** 3 good
**Contribution:** 3 good

**Summary:**

The paper proposes to extend the coordination graph framework to allow for non-linear mixing of agent payoffs/pairwise utilities, in a way similar to how the QMIX algorithm extends VDN by mean of mixing network. The problem of being able to solve the subsequent decentralized constraint optimization required to compute the optimal joint action is tackled by resorting to piece-wise neural network analysis, and nothing that it suffices to solve a linear DCOP for each possible slope configuration in order to find such optimal action. Moreover, an iterative version of this algorithm is proposed, possibly trading off optimality of the selected joint action with a reduced number of DCOPs to solve.

**Questions:**

- [Castellini et al., 2021](https://www.google.com/url?sa=t&rct=j&q=&esrc=s&source=web&cd=&ved=2ahUKEwj92tifi634AhU6RPEDHUIuBXUQFnoECBoQAQ&url=https%3A%2F%2Fresearch.tudelft.nl%2Ffiles%2F94310595%2FCastellini2021_Article_AnalysingFactorizationsOfActio.pdf&usg=AOvVaw0pSTZXIivp-1y3apJsQhMH), is such an extremely relevant citation here, showing the benefits of higher-order factorizations on tackling an exponential number of joint actions over both centralized controllers and independent learners.
- Your explanation of relative overgeneralization is a bit not-on-point (indeed, each agents is not supposed to know the actions of the others). Relative overgeneralization occurs when agents are pushed towards favouring a suboptimal behaviour because this gives, on average, a higher reward than the optimal coordinated behaviour. This happens because the concurrent learning process of the other agents shadows the optimal strategy with its exploration (equilibrium shadowing), rendering it less attractive than a suboptimal one.
- It may not be immediately clear how you decompose LeakyReLU$(\mathbf{o}_i)$ as $\mathbf{c}_i\circ \mathbf{o}_i$, where $c_i$ is simply the multiplication coefficient of the LeakyReLU (so either $\alpha$ or $1$). Therefore a reader may be a bit surprired of the definition of the slope configuration $\mathbf{c}\in\{\alpha,1\}^m$. Please clarify what $\mathbf{c}_i$ is and where do you get it.
- What do you mean exactly by a joint action $\mathbf{a}$ falling outside a given piece $\rho_k$ on line 164? Perhaps you mean that the $q_i,q_{ij}$s generated by the joint action $\mathbf{a}_k$ (I think these are $\mathbf{q}_k$ in your notation) generates a different slope configuration $\mathbf{c}^{m\not=k}$? This should be better clarified.
- What are the CG topologies for the different experiments for both your method and DCG? Why have you come up with such topologies?
- In Figure 2, when using NL-CG with an embedding of 4 units, is there any evidence that the iterative algorithm is performing well because it is indeed a good approximation and not because it is simply solving the constraint at every possible slope configuration (thus reducing to the enumeration algorithm in practice)? This should be shown somewhere, possibly in an Appendix if the space constraints are too limiting (although I would prefer to see such a figure rather than the very lengthy Algorithms 1, 2 and 3, that are not really adding much to the method explanation, that is already quite clear).
- In Figure 4, how do you guarantee that the iterative method is only solving the constrained problem for 16 slope configurations and not more?
- In Figure 5, what $Q_{tot}$ are you using as a reference? Are you using the one from NL-CG with enumeration? Or perhaps you are computing some sort of ground-truth Q-function? That should be specified for a reader to clearly understand and appreciate the results.

**Limitations:**

The possible limitations of the proposed method are actually never discussed in depth. The authors should make an effort and address this, trying to identify possible situations in which the proposed method could not perform as expected, and the reasons for this being the case.

**Strengths And Weaknesses:**

**Originality:** Up to the best of reviewer's knowledge, the idea of using a non-linear function to combine components in a coordination graph is novel, and so it is the algorithm proposed to identify the optimal joint action in such a non-linear representation.

**Quality:** The paper is in general technically sound, and the methodological claims are well supported by experimental results. The set of experiments and analysis proposed is good, and is really capable of addressing most of the concerns a reader may come up with. I only have some remarks for the authors, mainly concerning some more in-depth explanation of experimental details or to provide support of some claims. For example, it is never explained what CG topologies are used for the proposed experiments (also valid for the DCG baseline), while I think this is a relevant detail that should be reported. Or, where did you get the $Q_{tot}$ used in Figure 5? This is never explicitly explained, but is an important information to understand what the figure is representing.

Also, I have some small concerns about the performance of the iterative algorithm. For example, when using NL-CG with an embedding of 4 units in Figure 2, is there any evidence that the iterative algorithm is performing well because it is indeed a good approximation and not because it is simply solving the constraint at every possible slope configuration? This should be assessed before claiming that is is reducing the number of solved slope configurations as you are doing.

**Clarity:** The work is in general clearly written, and given the reader all of the required information to correctly understand both the proposed methodology and the experimental results, with just some minor exceptions (see **Questions** below).

Perhaps the explanation of the adaptation of Max-Sum to the non-linear DCOP setting may be a bit improved: what do you mean exactly by a joint action $\mathbf{a}$ falling outside a given piece $\rho_k$ on line 164? Perhaps you mean that the $q_i,q_{ij}$s generated by the joint action $\mathbf{a}_k$ (I think these are $\mathbf{q}_k$ in your notation) generates a different slope configuration $\mathbf{c}^{m\=k}$?

**Significance:** The presented idea is indeed significant, and advances our understanding of the field. With the great interest that both value-decomposition methods (with a and growing impact of higher-order decompositions)  and non-linear mixing architectures are gaining, this is a valuable contribution and could possibly allow for even better application of mixing techniques to coordination graph learning (that has proved indeed useful in learning better representations [Castellini et al., 2021](https://www.google.com/url?sa=t&rct=j&q=&esrc=s&source=web&cd=&ved=2ahUKEwj92tifi634AhU6RPEDHUIuBXUQFnoECBoQAQ&url=https%3A%2F%2Fresearch.tudelft.nl%2Ffiles%2F94310595%2FCastellini2021_Article_AnalysingFactorizationsOfActio.pdf&usg=AOvVaw0pSTZXIivp-1y3apJsQhMH).

---

> ### Author Response · Authors · 2022-08-02
> **Thanks for your review!**
>
>
> > **Question 5 & Quality 1** What are the CG topologies for the different experiments for both your method and DCG? Why have you come up with such topologies?
>
> - For both DCG and our method, we use the complete graphs for all experiments in the paper.
>
> - This is because complete graphs are the most basic topology and do not need to be pre-defined. Given that we aim to establish the basic concepts of non-linear CGs in this paper, complete graphs are a good choice.
>
> - Sparse or dynamic topologies may have some interactions (which are now still unclear) with non-linear CGs and may further improve the performance. We believe it is an interesting topic to be studied in further work.
>
> - Loopy graph topologies influence some analyses in the paper, we add a subsection at the end of the method section (line 216-225) to discuss the influence in detail.
>
> > **Quality 2** Is there any evidence that the iterative algorithm is performing well because it is indeed a good approximation and not because it is simply solving the constraint at every possible slope configuration?
>
> We provide additional experimental results showing the the iterative method is not enumerating all possible slope configurations. On task *Aloha*, with a mixing network of width 10, 12.756 slope configurations are visited on average out of all 1024 configuration. Moreover, 2.647 and 4.280 configurations are visited on average when the width is 3 and 4. This kind of approximation can often leads to nearly optimal results as in enumerating all configuartions.
>
> > **Question 1** Castellini et al., 2021, is such an extremely relevant citation here.
>
> Thanks for pointing out this important work. In the revised paper, we discuss Castellini et al., 2021 in the related work section.
>
> > **Question 2** Your explanation of relative overgeneralization is a bit not-on-point.
>
> We changed our explanation of relative overgeneralization to "relative overgeneralization embodies that, due to the concurrent learning and exploration of other agents, the employed utility function may not be able to express optimal decentralized policies and prefer suboptimal actions that give higher returns on average.". (l.29 - l.31)
>
> > **Question 3** It may not be immediately clear how you decompose LeakyReLU($\boldsymbol{o}_i$) as $\boldsymbol{c}_i \circ \boldsymbol{o}_i$.
>
> Thanks for this comments! We add clarification about what $\boldsymbol{c}$ is and how we get it at l.152.
>
> > **Question 4 & Clarity 1** what do you mean exactly by a joint action $\boldsymbol{a}$ falling outside a given piece $\rho_k$ on line 164? Perhaps you mean that the $\boldsymbol{q}$s generated by the joint action $\boldsymbol{a}$ generates a different slope configuration ?
>
> - Yes, the reviewer is correct, and we mean the joint action yields a different slope configuration.
>
> - We improve the presentation of this part by defining the *cell* $P_k$ of an affine function $\rho_k$ where $\boldsymbol{q}\in P_k$ actually yields $\boldsymbol{c}^k$ in a forward pass. Other parts of the method section are also modified, using this concept to eliminate possible ambiguity.
>
> > **Question 6** In Figure 2, when using NL-CG with an embedding of 4 units, is there any evidence that the iterative algorithm is performing well because it is indeed a good approximation and not because it is simply solving the constraint at every possible slope configuration (thus reducing to the enumeration algorithm in practice)?
>
> Yes. On task *Aloha* we count the average iterations that is needed before converging to a local optima, and the number is 4.28 for NL-CG with an embedding of 4 units. Moreover, 2.647 and 12.756 configurations are visited on average before converging with an embedding of 3 and 10 units, respectively.
>
> > **Question 7** In Figure 4, how do you guarantee that the iterative method is only solving the constrained problem for 16 slope configurations and not more?
>
> We guarantee this by directly stop iteration when $16$ slope configurations are visited.
>
> > **Question 8** In Figure 5, what $Q_{tot}$ are you using as a reference? Are you using the one from NL-CG with enumeration? Or perhaps you are computing some sort of ground-truth Q-function?
>
> Yes, the reference in Figure 5 is the $Q_{tot}$ found by NL-CG with enumeration. This is because this method has a global optimality guarantee.

---

> > ### Comment · Reviewer_Gt8B · 2022-08-03
> > **Response to the authors**
> >
> > The reviewer would like to thank the authors for their useful comments. Let me now try and discuss a bit on some of these (I generally acknowledge the corrections and clarifications that you applied straight away, without the need to explicitly mention these here):
> >
> > > - For both DCG and our method, we use the complete graphs for all experiments in the paper.
> > >
> > > - This is because complete graphs are the most basic topology and do not need to be pre-defined. Given that we aim to establish the basic concepts of non-linear CGs in this paper, complete graphs are a good choice.
> >
> > Fine, but this should be clearly stated in the paper to clarify this.
> >
> > > We provide additional experimental results showing the the iterative method is not enumerating all possible slope configurations.
> >
> > This is a nice addition and is definitely going to improve the overall quality of the work.
> >
> > > We guarantee this by directly stop iteration when $16$ slope configurations are visited.
> >
> > Again, please clearly state this.
> >
> > > Yes, the reference in Figure 5 is the $Q_{tot}$ found by NL-CG with enumeration. This is because this method has a global optimality guarantee.
> >
> > This puzzles me a bit. First of all, how can you say that your method has **global optimality** guarantees? This is a condition that is hard to attain in general, and the CG you are using (although non-linear) may be still an approximation of the true problem structure (that may indeed not be a decomposable one by its own properties). Finally, you are using neural networks, that are known to invalidate the convergence guarantees of most RL methods. How are these global optimality guarantees derived?
> >
> > Moreover, the fact that $Q_{tot}$ is taken from NL-CG, although with complete enumeration,makes the fact that the best results comes from NL-CG with iterative search a bit less interesting and surprising: the learned representation is probably the closest one to that used as a reference, so it is probable that its results are the best.
> >
> > Also, how can some $\mathbf{q}$-values be better than those of the NL-CG with complete enumeration then?

---

> > > ### Author Response · Authors · 2022-08-06
> > > **Thanks a lot for your comments!**
> > >
> > > Thanks very much for your feedback! We further modify our paper according to your comments.
> > >
> > > > This should be clearly stated in the paper to clarify this. (For both DCG and our method, we use the complete graphs for all experiments in the paper.)
> > >
> > > Thanks for this good suggestion. We go through our paper and state this point clearly in the beginning of the method section (line 123), the discussion subsection (line 216-225), and the experiment setup descriptions (line 267-268).
> > >
> > > > Please clearly state that we stop iteration when $n_{max}$ slope configurations are visited.
> > >
> > > We now state this explicitly in the experiment setup descriptions (line 270).
> > >
> > > > About Figure 5.
> > >
> > > - The usage of the term global optimality is sloppy. The reviewer is right that we cannot guarantee this due to possibly insufficient representational power and the usage of deep networks.
> > >
> > > - We now modify Figure 5 and use ground-truth Q values (averaged Monte Carlo returns) as the reference point as suggested by the reviewer. Specifically,
> > >
> > >   1. We compare the ground truth Q estimates of DCG and NL-CG (embed=3, iterative optimization, $n_{max}=4$). The result shows that NL-CG learns a policy with higher value.
> > >
> > >   2. We compare the $Q_{tot}$ value of our enumerative and iterative optimization methods. The result demonstrates that the local optimum solutions found by our iterative method with simulated annealing typically have the same value as the solutions found by the enumerative method.
> > >
> > >   3. We compare the $Q_{tot}$ values estimated by NL-CG (embed=3, iterative, $n_{max}=4$) against ground truth Q estimates. The estimation errors on all tested state-action pairs are less than 20\%.
> > >
> > > - We now explain why some q-values with iterative NL-CG are better than those of enumerative NL-CG. In practice, we use Max-Sum on individual linear pieces, which is an approximation algorithm. Examples like the following one can be constructed to demonstrate that, in such situation, it is possible that enumerative NL-CG may find a sub-optimal solution.
> > >
> > >   Assume that there are 4 linear pieces $\rho_{1}$, $\rho_{2}$, $\rho_{3}$, $\rho_{4}$ and the true maximum $Q_{tot}$ values on them are 10, 11, 12, 13. By applying Max-Sum on each linear piece we get actions $\boldsymbol a_1$, $\boldsymbol a_2$, $\boldsymbol a_3$, and $\boldsymbol a_4$, respectively. Since Max-Sum is not accurate, the obtained actions may not be local optimizer. Assume that the values of these actions are $\rho_{1}(\boldsymbol q(\boldsymbol a_1))=6$, $\rho_{2}(\boldsymbol q(\boldsymbol a_2))=9$, $\rho_{3}(\boldsymbol q(\boldsymbol a_3))=8$, and $\rho_{4}(\boldsymbol q(\boldsymbol a_4))=7$.
> > >
> > >   When we enumerate all the pieces, we will select $\rho_{2}$ and get the final $Q_{tot}$=9.
> > >
> > >   Let's consider a case where the iterative search stops at $\rho_{3}$ after $n_{max}$ iterations. Suppose that Max-Sum on $\rho_{3}$ returns $\boldsymbol a_3$ that actually falls in the cell of $\rho_{4}$, and $\rho_{4}(\boldsymbol q(\boldsymbol a_3))=13>9$. In this case, the solution found by iterative NL-CG is better than that by enumerative NL-CG.

---

> > > > ### Comment · Reviewer_Gt8B · 2022-08-07
> > > > **Response to the authors**
> > > >
> > > > I appreciate your explanations, the point is clearer now. I would like to thanks the authors, I do not have any further concern or doubt.

---

> > > > > ### Author Response · Authors · 2022-08-09
> > > > > **Thanks!**
> > > > >
> > > > > Thanks for the reviewer's work！

---

### Official Review · Reviewer_PvfL · 2022-06-30

**Rating:** 7
**Confidence:** 4
**Soundness:** 4 excellent
**Presentation:** 3 good
**Contribution:** 4 excellent

**Summary:**

This paper addressed the problem of cooperative multi-agent reinforcement learning by proposing an improvement over value decomposition method. They extend the concept of coordination graph to non-linear combination of value functions.

The core of the method relies on coordination graph. In coordination graphs, an edge is drawn between two interacting agent and a joint value function is learned through reinforcement learning. Given such graph, a globally optimal joint action can be computed to maximize a combination of the edge value functions. Traditionally, those value functions are summed and we can use algorithms such as max-plus to find the joint action. The author extend this process to non-linear combinations with deep network using LeakyReLU activations.

They extend the max-plus algorithm to this class of graphs by decomposing the problem as piecewise linear combinations. The first proposed algorithm has an exponential time complexity in the width of the mixing network due to enumerating all possible slope configuration but the authors propose an approximation which converge to a local optimum and rely on an annealing strategy to try to escape it.

The experiments compare the performance of the whole algorithm on the MACO benchmark against two baselines: QMIX and DCG.
A second set of experiments compare the optimality of the action selection only between the two propose algorithm,  DCG and a random baseline.


**Questions:**

1. Is there an application that would lead to similar dynamics as the problem illustrated in section 4? It would be useful to provide an example to further motivate why we would need the extra complexity.
2. The graph structure is barely discussed, in the original max-plus there are issues with graphs presenting cycles, how would your algorithm be affected?
3. Why are the NL-CG method starting at higher position than the other methods in figure 2?
4. The performance of QMIX is surprisingly low, why is that? It would have been useful to compare in another MARL where QMIX is not so bad e.g. starcraft.
5. In figure 5, how can DCG be compared with the method since they should use different Qtot (linear vs non linear)?


**Limitations:**

The authors do not explicitly mention any limitations of their method. I believe it would be valuable to add a paragraph. An obvious limitation to this family of method is that they require the graph structure to be known. Discussing different type of graph structure or sparsity would have been useful.

**Strengths And Weaknesses:**

The problem of cooperative multi-agent RL has many societal applications and there is still a lot of progress to be done in being able to solve it efficiently. This paper contributes towards this goal by extending state of the art techniques based on value decomposition methods.

Extending CGs to non-linear combination does intuitively increase the representation capability of the global value function, similarly to QMIX vs VDN, I believe the idea is sound. The author demonstrates this through the use of an example in section 4 and further validate it empirically.

Overall the paper is well written and quite easy to follow. The algorithms are sound and clearly explained.

When extending the max-plus algorithm to piecewise linear combination, they clearly explain the results and the derivation seems correct.

The weighted max-sum could be moved to appendix.

The plots from figure 5 and 6 are quite hard to follow, and the axes are not clear at all. Even if the author try to explain their meaning in the caption, it would be wise to choose another type of representation that is more intuitive for the reader. Furthermore I believe there are some unclarity on the correctness of this experiment regarding the choice of Qtot (see questions).

---

> ### Author Response · Authors · 2022-08-02
> **Thanks for your review!**
>
> > **Limitation 1** The weighted max-sum could be moved to appendix.
>
> As suggested by the reviewer, we move this algorithm to Appendix B of the revised paper.
>
> > **Question 1** Is there an application that would lead to similar dynamics as the problem illustrated in section 4?
>
> Yes. This task actually features relative overgeneralization. The actions of other agents may shadow the better choice (State 2B) with their exploration, rendering it less attractive than a worse choice (State 2A).
>
> This example shows that DCG cannot address some cases featuring relative overgeneralization.
>
> > **Question 2** The graph structure is barely discussed, in the original max-plus there are issues with graphs presenting cycles, how would your algorithm be affected?
>
> Thanks for this important question! We assume that max-plus has an error rate of $e$ in loopy graphs. From the empirical study in [Wang et al. 2022], $e$ is typically smaller than 5\%.
>
> - Lemma 1 is not affected because it is a property of LeakyReLU Networks.
>
> - For Lemma 2, the maximum of solutions found by message passing in all slope configurations is the global optimum with a probability of $1-e$. An error occurs when message passing cannot find the right solution on the piece where the global optimum is located.
>
> - Our iterative method may stop earlier when message passing returns a wrong solution located in the current cell. The probability of this situation is less than $e$. Thus we have at least a probability of $(1-e)^{n}$ ($n$ is the number of iterations) to find the piece where the local optimum is located, and the final probability of finding the local optimum is larger than $(1-e)^{n+1}$.
>
> In the revised paper, we add a paragraph discussing the influence of loopy graph structures at the end of the method section (line 216-225).
>
> > **Question 3** Why are the NL-CG method starting at higher position than the other methods in figure 2?
>
> - The first point is the performance after training with around 20K samples. NL-CG can already learn something using these samples.
>
> - We further show results on the predator-prey task in the revised paper. Similarly, our method requires very few (20K-30K) samples to achieve DCG's performance after converges.
>
> > **Question 4** The performance of QMIX is surprisingly low, why is that? It would have been useful to compare in another MARL where QMIX is not so bad e.g. starcraft.
>
> - MACO benchmark features tasks that require sophisticate coordination among agents. Not only QMIX, most fully decomposed value function methods (e.g., DICG in Figure 5) cannot perform well on these tasks.
>
> - On a super-hard scenario, MMM2, from the SMAC benchmark, our method still outperforms QMIX by a large margin.
>
>
> > **Question 5** In figure 5, how can DCG be compared with the method since they should use different Qtot (linear vs non linear)?
>
> Although DCG and NL-CG uses different network structures and optimization methods, they are learning under the same environments, and thus the same reward settings. The maximum expected accumulated rewards should be the same.
>
> > **Limitation** The authors do not explicitly mention any limitations of their method.
>
> - As suggested by the reviewer, a major limitation of our method is possible failure case in loopy graphs. We add related discussion in the revised paper.
>
> - Another limitation is that we only consider complete coordination graphs in this paper. The interaction between sparse topologies and non-linear mixing function is quite interesting but remains largely unknown.
>
> Reference:
>
> [Wang et al. 2022] Wang, T., Zeng, L., Dong, W., Yang, Q., Yu, Y. and Zhang, C., 2021, September. Context-Aware Sparse Deep Coordination Graphs. In International Conference on Learning Representations.

---

> > ### Comment · Reviewer_PvfL · 2022-08-03
> > **Thank you for the answer and the interesting work**
> >
> > Thank you for the clarifications, I am still a bit confused by the usefulness of figure 5. By comparing the estimated Qtot, how could you distinguish between the case where the method is indeed learning a policy with higher value, and the case where the method is simply over-estimating the value?
> > It seems that it lacks a ground truth value estimate.

---

> > > ### Author Response · Authors · 2022-08-06
> > > **Thanks very much for your comments! We modify Figure 5 to incorporate your suggestions.**
> > >
> > > Thanks a lot for your feedback and pointing out the problem of Figure 5.
> > >
> > > We modify Figure 5 in the revised paper and introduce ground truth Q value estimates (averaged Monte Carlo returns) as suggested by the reviewer.
> > >
> > > Specifically,
> > >
> > > 1. We compare the ground truth Q estimates of DCG and NL-CG (embed=3, iterative optimization, $n_{max}=4$). The result shows that NL-CG learns a policy with higher value.
> > >
> > > 2. We compare the $Q_{tot}$ value of our enumerative and iterative optimization methods. The result demonstrates that the local optimum solutions found by our iterative method with simulated annealing typically have the same value as the solutions found by the enumerative method.
> > >
> > > 3. We compare the $Q_{tot}$ values estimated by NL-CG (embed=3, iterative, $n_{max}=4$) against ground truth Q estimates. The estimation errors on all randomly selected state-action pairs are less than 20\%.

---

> > > ### Author Response · Authors · 2022-08-09
> > > **Thanks!**
> > >
> > > Thanks for the reviewer's work!

---

### Official Review · Reviewer_1Zfb · 2022-07-08

**Rating:** 7
**Confidence:** 5
**Soundness:** 4 excellent
**Presentation:** 3 good
**Contribution:** 4 excellent

**Summary:**

The paper introduces a convex mixing network, that is learned using a QMIX-style hyper-network, for coordination graphs. The authors prove that a maximum over the piecewise-linear parts of the mixing network corresponds to the global maximum and introduce an iterative optimization method for larger mixing networks that converges to a local maximum piecewise-linear solution. The resulting algorithms are compared with DCG on some MACO benchmark, and show impressive improvements on some. The authors also compared the enumeration over all parts with their iterative method in ablation studies.

**Questions:**

1) Is the wall-clock time of NL-CG simply DCG_time times num_iterations, or does the initialization with the optimal actions of previous iterations speed up the message passing?

2) Did you follow all the design decisions of the DCG implementation, or did you leave out some things (e.g. the state-dependend bias)?

3) Do you have an explanation why NL-CG performed so much better than DCG in Aloah and Sensor (but not in Gather)?


**Limitations:**

- A wallclock-time comparison to DCG would have been nice.
- It is unclear how well the algorithm scales to realistic applicatons with large action spaces, like the StarCraft2 experiments in the DCG paper.


**Strengths And Weaknesses:**

The reviewer liked the paper a lot, especially the theoretical part. Although the authors should better distinguish between the domain of a function and the inputs that "correspond" to a function (see detailed comments), the reviewer has checked the proofs thoroughly and believes they are correct. The authors should mention that their restrictions on the mixing network is almost identical to input-convex neural nets (ICNN, Amos et al., 2016). ICNN are a bit more general, but the theoretical analysis should still hold. In particular Lemma 1 looks as if it could have useful applications in other fields using ICNN.

Section 4 is not particularly illustrative, but the experimental experimental validation looks very well done and the results are very good on Aloha and Sensor. The biggest open question is scalability: a mixing network of size 10 might still be considered small. The authors could provide a better runtime comparison of their algorithm in comparison to DCG (see questions), and with number of network size and number of iterations of the approximate algorithm.

**Detailed comments**

- eq.4+5: as the mixing network $f_n$ depends on $s$, so must $Q_{tot}$
- l.132: "extended to other activation functions like ReLU" -- ReLU is already a special case of LeakyReLU with $\alpha=0$
- l.150: the outputs $o_i$ can easily be confused with the observations $o_i$
- l.162ff: you should differentiate between the domain of affine function $\rho_k$, which is the entire input space (otherwise one could not compute the output), and the inputs $q$ that "correspond" to $\rho_k$, that is, where the forward pass produces $c^k$. The two words are currently nowhere properly defined and used synonymous (e.g. Lemma 1 or l.172). Maybe define the set $q \in Q_k$ in which the forward pass yields $c^k$ to be precise.
- Lemma 1 is really interesting, but should be expressed more precisely: for which class of functions does it hold, and how the $\rho$ functions are defined. This will allow easier transfer to other fields. You can also remove the $s\neq r$ restriction, as for $s=r$ the inequality still holds.
- l. 169: it took the reviewer some time to believe the claim $h^r_{ij} \geq h^s_{ij}$. It would help to emphasize that the two slope configurations are identical, except for entry ${ij}$, and that the output $o_{ij}$ is therefore the same in both functions. A sentence about how the inequality holds might also help.
- l.184: the word "infeasible" is unclear here; you need to establish the terminology as suggested above
- l.200: as you mentioned later, using the condition $c_{real} = c_p$ can lead to loops. Why don't you just use $\rho_{real}(q(a_{real})) = \rho_p(q(a_p))$?
- l.217: the "reward is invariant to the identity" only for the second decision
- l.221ff: the example is very unclear. How do the 5 equations look like? Why do you set $q_i(s_{2B}, A)=0$ (doesn't this already used up the equation for 0 action B's)? All together the reviewer did not see a large benefit of Section 4.
- l.252: why 4 slopes? Shouldn't it be 3^2=8 slope configurations?
- l.252: it states that "$n_{max}=4$", but the Figure states $n_{max}=8$
- l.272f: it is unclear how you compare NL-CG with DCG in Figure 5. How does DCG get different entries on the x-axis?

**References**

Amos et al., 2016: Input Convex Neural Networks; https://arxiv.org/abs/1609.07152

---

> ### Author Response · Authors · 2022-08-02
> **Thanks a lot for your insightful and inspiring comments! We find most of them really helpful, deepening our understandings of non-linear coordination graphs.**
>
>
> > **Question 1 \& Limitation 1**
> >
> > Is the wall-clock time of NL-CG simply DCG\_time times num\_iterations, or does the initialization with the optimal actions of previous iterations speed up the message passing?
>
> - The initialization with the optimal actions of previous iterations speed up the message passing.
>
> - On task *Aloha*, we test the runtime of our method. With an embedding of 2, 4, and 10, our iterative method needs 2.647, 4.28, and 12.7 iterations on average to converge to local optimum. These iterations consumes 0.12 $ms$, 0.15 $ms$, and 0.3 $ms$ together. So the average time for one Max-Sum drops from 0.0453 $ms$ to 0.0350 $ms$, and further to 0.0236 $ms$ when the iteration number increases.
>
> > **Question 2**
> >
> > Did you follow all the design decisions of the DCG implementation, or did you leave out some things (e.g. the state-dependend bias)?
>
> We follow all the design decisions of DCG. Both NL-CG and DCG are tested with the state-dependent bias.
>
> > **Limitation 2**
> >
> > It is unclear how well the algorithm scales to realistic applicatons with large action spaces, like the StarCraft2 experiments in the DCG paper.
>
> - In the revised paper, we test our method on predator-prey tasks and SMAC. The results are shown in Figure 5.
>
> - On predator-prey, our method requires very few (20K-30K) samples to achieve DCG's performance after converges.
>
> - On a super-hard scenario, MMM2, from the SMAC benchmark, our method achieves a win rate of 80\% after 3M training steps, while DCG achieves around 60\%.
>
> These results demonstrate the effectiveness of our method in complex scenarios.

---

> > ### Comment · Reviewer_1Zfb · 2022-08-06
> > **Thanks**
> >
> > Thanks to the authors for their answers.

---

> > > ### Author Response · Authors · 2022-08-09
> > > **Thanks!**
> > >
> > > Thanks for the reviewer's work.

---

> ### Author Response · Authors · 2022-08-02
> **Thanks a lot for your insightful and inspiring comments! We find most of them really helpful, deepening our understandings of non-linear coordination graphs.**
>
>
> > **Strengths And Weaknesses 1**
> >
> > The authors should better distinguish between the domain of a function and the inputs that "correspond" to a function.
>
> - As suggested by the reviewer, we differentiate these two concepts by defining the *cell* $P_k$ of an affine function $\rho_k$ where $\boldsymbol{q}\in P_k$ yields $\boldsymbol{c}^k$ in a forward pass. The presentation of the method section is also modified, using this concept to eliminate possible ambiguity.
>
> > **Strengths And Weaknesses 2**
> >
> > The authors should mention that their restrictions on the mixing network is almost identical to input-convex neural nets
>
> - Thank you very much for pointing out the relationship to ICNN. We mention ICNN at l.134 of the revised paper and discuss how Proposition 1 in the ICNN paper relates to the representational power of our mixing network.
>
> > **Detailed comments 1** eq.4+5: as the mixing network depends on $s$, so must be $Q_{tot}(s,a)$.
>
> - We changed $Q_{tot}(\boldsymbol\tau, \boldsymbol a)$ in Eq. 4 and 5 to $Q_{tot}(s, \boldsymbol a)$.
>
> > **Detailed comments 2** l.132: "extended to other activation functions like ReLU" -- ReLU is already a special case of LeakyReLU
>
> - We removed "extended to other activation functions like ReLU".
>
> > **Detailed comments 3** l.150: the outputs $o_i$ can easily be confused with the observations $o_i$.
>
> - Throughout the revised paper, we use a different notation $\boldsymbol{z}_i$ for the output to improve our presentation.
>
>
> > **Detailed comments 4** l.162ff: you should differentiate between the domain of affine function $\rho_k$ and the inputs that "correspond" to $\rho_k$.
>
> - See Strengths and Weaknesses 1.
>
> > **Detailed comments 5** Lemma 1 is should be expressed more precisely.
>
> - We improved the expression of Lemma 1 by specifying the function class for which it holds, giving the definition of linear pieces, and removing the restriction $r\ne s$.
>
> > **Detailed comments 6** l. 169: it took the reviewer some time to believe the claim $h^r_{ij}\ge h^s_{ij}$.
>
> - To make why $h^r_{ij}\ge h^s_{ij}$ more clear, we added that the output $o_{ij}$ (in the revised notation, $z_{ij}$) is the same and explicitly showed the comparison between $h^r_{ij}$ and $h^s_{ij}$ by expanding them in a new equation 6.
>
> > **Detailed comments 7** l.184: the word "infeasible" is unclear here.
>
> - We made the meaning of "infeasible" clear by incorporating the terminology *cell* defined above.
>
> > **Detailed comments 8** l.200: as you mentioned later, using the condition $c_{real} = c_p$ can lead to loops. Why don't you just use $\rho_{real}(q(a_{real})) = \rho_{p}(q(a_{p}))$?
>
> - This is because, in practice, we run Max-Sum in each piece, which may be inaccurate in loopy graphs, and exact equality between Q values is a strict condition.
>
> > **Detailed comments 9** l.217: the "reward is invariant to the identity" only for the second decision.
>
> - l.217: This also holds for the first decision, because the reward depends only on the number of agents that take Action B for both decisions.
>
> > **Detailed comments 10** l.221ff: the example is very unclear. How do the 5 equations look like? Why do you set $q_i(s_{2B}, A)=0$? All together the reviewer did not see a large benefit of Section 4.
>
> - We specify the system of equations in Appendix C of the revised paper. This system has no solution because the rank of its augmented matrix is larger than its coefficient matrix.
>
> > **Detailed comments 11** l.252: why 4 slopes?
>
> - l.252: Sorry this is a typo, there are 8 slope configurations to enumerate.
>
> > **Detailed comments 12** l.252: it states that $n_{max}=4$, but the Figure states $n_{max}=8$.
>
> - l.252: we updated our paper and $n_{max}$ should be 8.
>
> > **Detailed comments 13** l.272f: it is unclear how you compare NL-CG with DCG in Figure 5. How does DCG get different entries on the x-axis?
>
> - l.272: One point in Figure 5 corresponds to one timestep (and thus a graph instance) in the game. Its x-value is $Q_{tot}$ of the solution found by DCG, while its y-value is $Q_{tot}$ of the solution found by NL-CG.

---

### Official Review · Reviewer_W1Zi · 2022-07-11

**Rating:** 7
**Confidence:** 3
**Soundness:** 3 good
**Presentation:** 3 good
**Contribution:** 3 good

**Summary:**

This paper investigates the problem of learning non-linear coordination graphs in multi-agent reinforcement learning. It proposes the first non-linear coordination graph by extending CG value decomposition beyond the linear case. It proposes the weighted max-sum algorithm to solve the greedy-action selection problem in the non-linear coordination graph.

**Questions:**

I have the following questions:

1. In lines 112-113, why is a DNN with piece-wise linear (PWL) activation functions (e.g. ReLU, LeakyReLU, PReLU) is equivalent to a PWL function? Did it motivate you to investigate the problem of the non-linear coordination graph?

2. In lines 141-142, when the mixing network is non-linear, maximizing Q_tot is NP-hard. Can you elaborate more? As far as I know, deep networks have capacities to learn good models.

3. Can you highlight your contributions in Alg. 1, 2 and 3?


**Limitations:**

Please see the above comments.

**Strengths And Weaknesses:**

Strengths:

This paper investigates a very important problem in coordination graphs. Previous methods focus on the linear case, which has limited representation capacity on Q values. This paper proposes a novel method by using a mixing network, the active function is LeakyReLU.


Weaknesses:

1. Using a mixing network to represent the coordination graph is not new. Previous work [1] uses GNN, which can seem like a mixing network, and ReLU or LeakyReLU can also be used in GNN.

2. The experiments are not strong, CASEC [2] is not compared.

3. More scenarios should be tested. As this paper considers the representation capacity issue, more experiments in complex scenarios, for example, the predator-prey task and SMAC in [3], should be conducted to show the merit of the method.

4. Some curves are not fully shown. For example, in Fig 3, the curves of NL-CG are not fully shown.

5. This paper is hard for me to follow. The writing can be improved.

[1] Deep Implicit Coordination Graphs for Multi-agent Reinforcement Learning

[2] CONTEXT-AWARE SPARSE COORDINATION GRAPHS

[3] Deep Coordination Graphs

---

> ### Author Response · Authors · 2022-08-02
> **Thanks for your review. We provide new experimental results and clarification to answer your questions.**
>
> Reference:
>
> [Chu et al. 2018] Lingyang Chu, Xia Hu, Juhua Hu, Lanjun Wang, and Jian Pei. Exact and consistent interpretation for piecewise linear neural networks: A closed form solution. In Proceedings of the 24th ACM SIGKDD International Conference on Knowledge Discovery & Data Mining, pages 1244–1253, 2018.

---

> ### Author Response · Authors · 2022-08-02
> **Thanks for your review. We provide new experimental results and clarification to answer your questions.**
>
> > **Weakness 1**
> >
> > Using a mixing network to represent the coordination graph is not new. Previous work [1] uses GNN, which can seem like a mixing network, and ReLU or LeakyReLU can also be used in GNN.
>
> - The concept of *coordination graphs* is different in [1] and our work. [1] mixes **individual** utility functions $q_i(a_i)$. In contrast, we additionally mixes **pairwise** payoff functions, which represent a higher order decomposition of the global value function.
>
> - The existence of pairwise payoff functions makes a big difference, because the value-maximizing actions of local utility functions are no longer global value-maximizers, and we have to develop new DCOP algorithms for greedy action selection.
>
> - The above claim can be supported by experimental results in Figure 5. We can see that DICG has similar performance to QMIX on task predator-prey and MMM2.
>
> > **Weakness 2**
> >
> > The experiments are not strong, CASEC [2] is not compared.
>
> - CASEC is orthogonal to our work. Our non-linear coordination graphs can also use the technique in CASEC to exploit the benefits of sparse graph topologies, which will further improve the performance of our methods.
>
> - This paper aims to develop basic and general concepts/properties of non-linear coordination graphs. In our humble opinion, the interaction between sparse (or any other) topologies and non-linear mixing functions deserves in-depth studies in the future work.
>
> > **Weakness 3**
> >
> > More scenarios should be tested. As this paper considers the representation capacity issue, more experiments in complex scenarios, for example, the predator-prey task and SMAC in [3], should be conducted to show the merit of the method.
>
> As suggested by the reviewer, we test our method on predator-prey tasks and SMAC. The results are shown in Figure 5 of the revised paper.
>
> - On predator-prey, our method requires very few (20K-30K) samples to achieve DCG's performance after converges.
>
> - On a super-hard scenario, MMM2, from the SMAC benchmark, our method achieves a win rate of 80\% after 3M training steps, while DCG achieves around 60\%.
>
> These results demonstrate the effectiveness of our method in complex scenarios.
>
> > **Weakness 4**
> >
> > Some curves are not fully shown. For example, in Fig 3, the curves of NL-CG are not fully shown.
>
> - In Fig. 3, we didn't show the full curves of the enumerating method with a mixing network of width 10. This is because we can draw a conclusion from the partial results -- a wide mixing network can help improve performance in some tasks (e.g., Hallway).
>
> - Besides performance, another dimension to compare the two versions of our method is time complexity. Running these incomplete curves is extremely time-consuming because they are enumerating all possible slope configurations. For comparison, with a mixing network of the same width, our iterative method runs much faster (Figure 4). Therefore, these incomplete results are in line with our motivation to develop the iterative optimization method.
>
> > **Question 1**
> >
> > In lines 112-113, why is a DNN with piece-wise linear (PWL) activation functions (e.g. ReLU, LeakyReLU, PReLU) is equivalent to a PWL function? Did it motivate you to investigate the problem of the non-linear coordination graph?
>
> The property of DNNs with piece-wise linear activation functions is well studied. We refer to [Chu et al. 2018] for detailed discussion. Our method is based on this property, which indeed provides an opportunity of extending coordination graphs to the non-linear case.
>
> > **Question 2**
> >
> > In lines 141-142, when the mixing network is non-linear, maximizing $Q_tot$ is NP-hard. Can you elaborate more? As far as I know, deep networks have capacities to learn good models.
>
> - $Q_tot$ is defined over the space of joint actions. When the mixing network is non-linear, to maximize $Q_tot$, one needs to enumerate all joint actions. The number of joint actions grows exponentially with the number of agents, and thus the problem is NP-hard.
>
> - Deep networks can learn good models, but the problem here is non-convex optimization problem over the *input* (instead of parameters) of a network in a exponentially growing space.
>
> > **Question 3**
> >
> > Can you highlight your contributions in Alg. 1, 2 and 3?
>
> As stated in the answer of the previous question, maximizing $Q_tot$ with a non-linear mixing network needs an enumeration over a space growing exponentially with the number of agents. Fortunately, we find that if the mixing network has a specific feature, i.e., if they use ReLU or LeakyReLU activation, the problem can be solved efficiently by two algorithms (Alg. 2 and 3). Our contribution is the procedure of Alg. 2 and 3. Alg. 1 is a sub-module for implementing Alg. 2 and 3, which extends the classic Max-Sum algorithm to weighted cases.

---

> ### Comment · Reviewer_W1Zi · 2022-08-07
> **Response to the Authors**
>
> Dear authors,
>
> Thanks for the authors' hard work on the responses. After reading the authors' response and revised paper. Most of my concerns are addressed. I raised the score.

---

> > ### Author Response · Authors · 2022-08-09
> > **Thanks!**
> >
> > Thanks for the reviewer's work!

---

### Meta-Review · Area_Chair_2hnu · 2022-08-26

**Recommendation:** Accept
**Confidence:** Certain

**Metareview:**

This paper is a very clear accept. The reviews had only minor quibbles, which I trust the authors will address in their final version.

**Award:**

No

---

### Decision · Program_Chairs · 2022-09-14

Accept